***Prog Neurobiol.** Author manuscript; available in PMC 2026 March 28.*

# Divergence of cortical neurophysiology across different neurodegenerative disorders compared to healthy ageing

**Michael Trubshaw**[a,b,1], **Oliver Kohl**[b,1], **Chetan Gohil**[b,1], **Mats W.J. van Es**[b], **Andrew J. Quinn**[b,c], **Katie Yoganathan**[a,b], **Evan Edmond**[a,b], **Malcolm Proudfoot**[a,b], **Nahid Zokaei**[b], **Vanessa Raymont**[d], **Jemma Pitt**[b], **Tony Thayanandan**[b], **Alexander G. Thompson**[a], **Kevin Talbot**[a], **Michele T. Hu**[a], **Marlou Nadine Perquin**[e], **Ece Kocagoncu**[e], **James B. Rowe**[e,f], **Mark W. Woolrich**[b], **Anna C. Nobre**[b,g,*,2], **Martin R. Turner**[a,b,**,2]

[a]Nuffield Department of Clinical Neurosciences, University of Oxford, Oxford, OX3 9DU, UK.

[b]Oxford Centre for Human Brain Activity, University of Oxford, Oxford, OX3 7JX, UK.

[c]Centre for Human Brain Health, School of Psychology, University of Birmingham, Birmingham, B15 2TT, UK.

[d]Department of Psychiatry, University of Oxford, Oxford, OX3 7JX, UK.

[e]Medical Research Council Cognition and Brain Sciences Unit, University of Cambridge, Cambridge, CB2 7EF, UK.

[f]Department of Clinical Neurosciences and Cambridge University Hospitals NHS Trust, University of Cambridge, Cambridge, CB2 0SP, UK.

[g]Wu Tsai Institute and Department of Psychology, Yale University, New Haven, CT, USA.

[*]Correspondence to: 100 College Street, New Haven, Connecticut CT 06510, USA. [**]Correspondence to: West Wing Level 6, John Radcliffe Hospital, Oxford OX3 9DU, UK. kia.nobre@yale.edu (A.C. Nobre), martin.turner@ndcn.ox.ac.uk (M.R. Turner).
[1]These authors contributed equally
[2]These authors contributed equally

**CRediT authorship contribution statement**
**Woolrich Mark W:** Writing – review & editing, Validation, Supervision, Software, Methodology, Investigation, Funding acquisition, Conceptualization. **Malcolm Proudfoot:** Writing – review & editing, Investigation, Data curation. **Nobre Anna C:** Writing – review & editing, Validation, Supervision, Methodology, Investigation, Funding acquisition, Conceptualization. **Nahid Zokaei:** Writing – review & editing, Investigation, Data curation. **Vanessa Raymont:** Writing – review & editing, Investigation, Data curation. **Hu Michelle T:** Writing – review & editing, Supervision, Investigation, Conceptualization. **Van Es Mats:** Writing – review & editing, Supervision, Project administration, Data curation. **Perquin Marlou:** Writing – review & editing, Investigation, Conceptualization. **Quinn Andrew J:** Writing – review & editing, Software, Investigation, Data curation. **Ece Kocagoncu:** Writing – review & editing, Investigation, Conceptualization. **Katie Yoganathan:** Writing – review & editing, Validation, Investigation, Data curation. **Rowe James B:** Writing – review & editing, Supervision, Investigation, Funding acquisition, Conceptualization. **Evan Edmond:** Writing – review & editing, Investigation, Data curation. **Jemma Pitt:** Writing – review & editing, Investigation, Data curation. **Tony Thayanandan:** Writing – review & editing, Investigation, Data curation. **Michael Trubshaw:** Writing – review & editing, Writing – original draft, Methodology, Investigation, Formal analysis, Conceptualization. **Thompson Alexander G:** Writing – review & editing, Investigation, Data curation. **Oliver Kohl:** Writing – review & editing, Writing – original draft, Methodology, Investigation, Formal analysis, Data curation, Conceptualization. **Kevin Talbot:** Writing – review & editing, Investigation, Conceptualization. **Chetan Gohil:** Writing – review & editing, Writing – original draft, Methodology, Investigation, Formal analysis, Data curation, Conceptualization. **Martin R.Turner:** Writing – review & editing, Supervision, Methodology, Investigation, Funding acquisition, Conceptualization.

**Declaration of Competing Interest**
None

## Abstract

Neurodegenerative diseases involve disruption of healthy brain network communication occurring before the emergence of symptoms. Magnetoencephalography (MEG) is sensitive to the magnetic fields generated by cortical neuronal activity, and is the most spatio-temporally accurate method of directly assessing neuronal activity non-invasively. We used MEG to directly compare three neurodegenerative disorders with a large healthy cohort to characterise patterns of activity deviating from healthy ageing.

Task-free MEG recordings were acquired from patients with Alzheimer's disease (AD, n = 29), Parkinson's disease (PD, n = 25), amyotrophic lateral sclerosis (ALS, n = 33) and healthy controls (HC, n = 191). Healthy ageing trajectories for metrics including spectral power (local neuronal recruitment), connectivity (long-range communication), 1/f exponent (power spectrum slope, which may reflect inhibition), and oscillatory speed were extracted. These metrics were compared pairwise between HC and patient groups, controlling for age and sex.

The modelled trajectories of healthy ageing included increasing beta power and oscillatory speed, with reduced power spectrum slope. PD, AD, and ALS groups all showed reductions in beta power and slowing of oscillatory activity compared to matched HC. In AD, older patients showed lower beta power compared with younger patients. Compared with matched HC, the power spectrum slope was uniquely reduced in ALS, in contrast to the increase seen in PD and AD. Gamma connectivity increased in AD and ALS.

MEG has unique potential as a source of biomarkers that might be used to detect deviation from healthy ageing if applied at an earlier presymptomatic stage of neurodegeneration than current tools permit. It might also provide outcome measures for prevention trials.

### Keywords

Magnetoencephalography; Neurodegeneration; Ageing; Biomarker; Networks

## 1 Introduction

The personal, societal, and global burden of neurodegenerative diseases is enormous and growing. Neurodegeneration broadly describes a process of neuronal dysfunction followed by loss of network integrity, leading to progressive impairment of brain and spinal cord function, including cognition, sensory processing and motor outputs (Azam et al., 2021). Ageing is the most consistent risk factor for neurodegenerative diseases (Azam et al., 2021; Keshavarz et al., 2023; Mir et al., 2023) and the context in which they develop. Many of the pathological changes seen in neurodegeneration, such as atrophy, neuroinflammation, disruption to proteostasis, and mitochondrial dysfunction, also occur to some extent in healthy ageing (Keshavarz et al., 2023; Turrini et al., 2023). Biomarkers of neurodegeneration have thus far proven elusive, but their development would undoubtedly facilitate more rapid attainment of disease-modifying therapeutics (Hansson, 2021a; Wilson et al., 2023). If these are to be deployed pre-symptomatically, then identifying deviation from healthy ageing and signatures specific to disease phenotypes will be essential.

Pathological processes underlying neurodegenerative diseases predominantly impact brain regions and networks in selective patterns (Jucker and Walker, 2013; Sami et al., 2018), resulting in specific pathological trajectories. Three common but distinct neurodegenerative diseases are Alzheimer's Disease (AD), Parkinson's Disease (PD), and amyotrophic lateral sclerosis (ALS). AD affects cognition, memory, and behaviour and has a global prevalence of 1.7 % (Gustavsson et al., 2023). Its peak incidence is between 80 and 90 years old (Masters et al., 2015; Wattmo et al., 2014). Mean adjusted survival from symptom onset is 8.8 years (Wattmo et al., 2014). AD is associated with cerebral extracellular insoluble amyloid-β deposition and the formation of intraneuronal tau neurofibrillary tangles (Masters et al., 2015; Wattmo et al., 2014). PD is defined by bradykinesia, tremor, rigidity, and postural instability with a prevalence of 0.3 % and a peak incidence age of 70–80 years (Pringsheim et al., 2014). Mean adjusted survival is 14.6 years. PD is associated with neuronal aggregates of misfolded α-synuclein (Golbe and Leyton, 2018; Jankovic and Tan, 2020). ALS involves a loss of motor system integrity, resulting in progressive lower and upper motor neuron weakness. ALS has a global prevalence of 0.04 %, a peak incidence age of 70–74 years old with a mean survival of less than 3 years from symptom onset (Logroscino et al., 2010; Xu et al., 2020). It is associated with neuronal and glial cytoplasmic aggregates of a 43 kDa transactive response DNA binding protein (TDP-43) and is part of a clinicopathological spectrum with frontotemporal dementia (Byrne et al., 2013; Kiernan et al., 2011). See Table 1 for a summary of AD, PD and ALS.

AD, PD, and ALS share a common theme of pathological protein aggregation as the molecular substrate for nervous system dysfunction (Wilson et al., 2023). Aggregation likely begins many years prior to symptom onset (Peng et al., 2020). Measuring the existence and distribution of early proteinopathy is an obvious biomarker target; however, direct and reliable correlations with symptomatic disease expression are not yet possible (Alcolea et al., 2023). Among the earliest correlates of this proteinopathy might be changes in brain function at the network level (Perovnik et al., 2023). MRI has played a major role in characterising patterns of structural and functional dysfunction in various neurodegenerative disorders (Hansson, 2021b). Cortical and subcortical atrophy, white matter tract degradation, and hypometabolism have been identified as key features of neurodegeneration (Sarasso et al., 2021; Talwar et al., 2021; Zejlon et al., 2022). However, these may be relatively downstream and late pathological changes. Functional MRI (based on a blood oxygenation level-dependent, BOLD, activation surrogate) is limited by its poor temporal resolution and is insensitive to changes in neuronal dynamics reflecting interacting ultra-fast functional brain networks characterising healthy brain function (Chandrasekar et al., 2025; Gohil et al., 2022).

Magnetoencephalography (MEG) offers a promising complementary approach and is well placed to detect cortical network activity non-invasively, in both states of synaptic health and neurodegeneration (Cope et al., 2022; Kocagoncu et al., 2022, 2020; Price et al., 2017; Sami et al., 2018; Vaghari et al., 2022). MEG sensitively and directly measures magnetic fields generated by cortical neuronal populations, circum-venting confounding haemodynamic factors (Gross, 2019). Its high density of sensors and the measurement of magnetic, rather than electrical (as in electroencephalography) signals, gives MEG high spatial resolution and very high temporal resolution (Cicmil et al., 2014). This combination allows MEG

to characterise local neuronal activity and, more broadly, whole-brain networks (Gohil et al., 2022), which are likely disrupted at very early stages of disease, prior to the onset of symptoms (Dukic et al., 2025; Gross, 2019; Lanskey et al., 2022; Trub-shaw et al., 2025a).

To date most studies have focused on comparing a single neurode-generative condition to healthy controls. The few available studies comparing across neurodegenerative diseases have provided valuable insights into the meaning of differential spectral properties between different diseases, pinpointing their origin to distinct patterns of neuronal microstructure disruption (Cope et al., 2022; Sami et al., 2018). Direct comparison of the cortical neurophysiology of groups of individuals with a diagnosis of AD, PD, or ALS, alongside a healthy cohort, offers the potential to identify key features linked to divergence from healthy network function, and the specific signatures of three distinct neurodegenerative phenotypes.

## 2 Methods

### 2.1 Study design

Participants were recruited to observational case-control studies at two UK centres – the Oxford Centre for Human Brain Activity at the University of Oxford (AD, PD, ALS, HC) and the MRC Cognition and Brain Sciences Unit at the University of Cambridge (AD, HC). To investigate the effects of healthy ageing on MEG metrics, in addition to the age- and sex-matched HC cohort (recruited via friends and relatives of patients through the OxDARE registry (NIHR Oxford Health Biomedical Research Centre, 2025)), a further cohort of age- and sex-matched younger HC was included from the publicly available MEGUK dataset (MEG UKI, 2024). All those with AD, ALS and PD were recruited from local clinical services and clinically diagnosed with sporadic disease by a neurologist or old age psychiatrist. PD patients were asked to withdraw their medication at 7 pm the night before the study. ALS and PD patients with clinical dementia were excluded. Full inclusion and exclusion criteria for each disease group can be found in Supplementary Table 1. Ethical approval was granted by National Research Ethics Service Committees (12/SC/0650 (PD), 14/SC/0083 (ALS), 17/SC/0277 (ALS), 18/EE/0042 (AD)).

### 2.2 MEG acquisition

Prior to scanning, head shape was recorded using a Polhemus 3D tracking system relative to three fiducial points on the nasion and two preauricular landmarks. Five head-position-indicator (HPI) coils were placed on the participant's nasion, posterior auricular, and bilateral supraorbital regions. The locations of the HPI coils and fiducials were digitised using the tracking system to define a participant-specific cartesian head coordinate system.

The MEG recordings were obtained in a magnetically shielded room using a 306-sensor (204 first-order planar gradiometers, 102 magne-tometers) MEGIN system (VectorView and TRUIX Neo) at the Oxford Centre for Human Brain Activity. The MEG signal was sampled at 1000 Hz, with a high-pass filter at 0.1 Hz and a low-pass anti-aliasing filter at 330 Hz. During scanning, participants underwent five to ten minutes of task-free MEG with their eyes open. They were instructed to fixate visually on a cross displayed 120 cm in front of them. Locations of the HPI coils were continuously monitored in scanner space.

After MEG scanning, participants underwent a structural T1-weighted MRI for MEG co-registration. Participants underwent a T1-weighted structural MRI scan with a Siemens Trio 3 T (settings: 3-dimensional, whole-brain, magnetization-prepared rapid-acquisition gradient echo sequence, repetition time = 1900 ms, echo time = 4.7 ms, flip angle 8°, 1 mm isotropic resolution, 7 min acquisition time) within one month for MEG co-registration.

## 2.3　Data processing

The raw data were preprocessed and analysed for each participant in the same way. The Oxford Centre for Human Brain Activity (OHBA) Software Library (osl-ephys) v0.6.0 Python package (built on MNE Python (Gramfort, 2013)) was used for data processing and metric estimation (Quinn et al., 2023; van Es et al., 2024). Denoising, bad channel detection, and head movement correction were performed using the temporal signal space separation algorithm (tSSS) provided in the MaxFilter software package v2.2 (Andrew J Quinn, 2021). A bandpass filter was applied between 0.5 and 125 Hz. A notch filter at 50 Hz and 100 Hz removed line noise. The data were resampled to 250 Hz. Bad segments were removed using osl-ephys' automated algorithm based on detecting abnormally high windowed variance (with default settings). MNE's signal-space projection algorithm was used to further denoise artefacts visually by consensus (MT, OK, CG) (Larson et al., 2024; Uusitalo and Ilmoniemi, 1997). osl-ephys' RHINO tool was used for co-registration (using each participant's fiducials, >100 scalp head shape points and their individual structural MRI) and single shell forward modelling. A final bandpass filter was applied between 1–80 Hz prior to source localisation, which allowed studying oscillatory activity within this range. Data were beamformed using a unit-noise-gain LCMV beamformer to a regular 8×8×8 mm$^3$ dipole grid using a data covariance matrix regularised to a rank of 60, with dipole orientations calculated by maximising each dipole's power (Woolrich et al., 2011). Note, the LCMV beamformer used in this work normalises the beamformer weights to produce dipoles with unit variance. This corrects for the depth bias that occurs in MEG source reconstruction (Tait et al., 2021) and also means the estimated source (dipole) time course is unitless.

The osl-ephys 'spatial basis' method was used to parcellate source localised data using the Glasser52 parcellation (Kohl et al., 2023). Each parcel time-course was estimated as the first principal component over all voxels forming a parcel (Colclough et al., 2015).

## 2.4　Metric calculation

Static spectral power, a measure of local neuronal activity, was calculated using Welch's method with a Hann window of 2 s from each *z*-transformed (standardised) parcel time course. Power spectral densities (PSDs) were extracted for each region (parcel) individually (see Fig. 1i for an example of a PSD). Power was estimated across six canonical frequency bands: delta (1–4 Hz), theta (4–7 Hz), alpha (7–13 Hz), beta (13–30 Hz), low-gamma (30–48 Hz), and high-gamma (52–80 Hz).

Spectral shape metrics included the 1/f exponent (calculated as the slope of the aperiodic component of the PSD) and oscillatory speed (calculated as the centre of energy of the periodic component of the PSD) for each region individually. The FOOOF algorithm was

used to parametrise each PSD between 1 and 70 Hz (higher frequencies were excluded to reduce the spectral plateau bias effect (Gerster et al., 2022)) and extract the aperiodic exponent and the periodic components (de-FOOOFed spectrum) (Donoghue et al., 2020). Settings included peak width limit: 0.5–12.0, maximum number of peaks: ∞, minimum peak height: 0.05, peak threshold: 2.0, and aperiodic mode: fixed. The aperiodic component slope was taken as a measure of signal complexity (Donoghue et al., 2020; Gerster et al., 2022). This was done for each region (parcel) individually.

As an estimation of 'oscillatory speed', the centre of energy (CoE) was calculated in each region (Krösche et al., 2023a). The de-FOOOFed spectrum (i.e. the power spectrum containing only the periodic components) was first extracted and then the frequency at which there was a balance of spectral power between lower and higher frequencies was calculated. This was done for each region (parcel) individually. The CoE corresponds to the average frequency of oscillations, where higher/lower values reflect faster/slower oscillations on average in the data. 'Oscillatory slowing' is represented by a shift of the CoE to the left (decrease), and an oscillatory acceleration by a shift to the right (increase) (Fig. 1i-iii). CoE is a broad measure of oscillatory speed and may be affected by different spectral phenomena including alpha or beta peak shift and change in alpha or beta peak amplitude (Fig. 1iii) (Krösche et al., 2023a). To investigate the cause behind any shift in CoE, a supplementary 'peak analysis' was conducted by extracting alpha and beta peak frequencies and power (over and above the 1/f exponent) using the FOOOF algorithm for each participant in each parcel. Where no peak was identified, the mean value across all participants was imputed. When multiple peaks were identified, the peak with the highest power was selected (Babiloni et al., 2017). To assess potential group differences in peak detectability, the number of missing alpha and beta peaks were compared between groups.

To quantify long-range communication, connectivity was estimated by calculating the AEC matrices for each frequency band separately from the *z*-transformed time-courses. As a measure of global connectivity, the mean connectivity across one axis of the AEC matrix was taken to estimate each region's AEC value with every other region. Note, because the AEC is the Pearson correlation between amplitude time courses, it is not sensitive to the amplitudes themselves, i.e., it is an amplitude-normalised measure of connectivity. This means regions with higher power (amplitude) do not automatically have higher connectivity. A summary of how each metric was calculated is shown in Fig. 1.

## 2.5 Statistical analysis

Python (version 3.8.15), GLMTools (version 0.2.0), and SciPy (version 1.10.0) packages were used for all statistical analyses (Andrew J Quinn, 2023; Quinn et al., 2024; Virtanen et al., 2020). To provide a reference for how power, spectral shape, and connectivity change in response to healthy ageing, data was collated from the age-matched HC group (n = 114) and an additional cohort of young HCs (n = 77). Each metric of cortical activity (power, 1/f exponent, oscillatory speed (CoE), and connectivity) was summarised using their own separate fit to a General Linear Model (GLM), which models variability over subjects using regressors for age, sex, and scanner type. Age (as a continuous variable) was used as the

predictor to show the effect of age on the extracted metrics (see Supplementary Figure 1 for design matrix).

To compare disease groups and healthy controls pairwise (PD-HC, ALS-HC, AD-HC, PD-ALS, ALS-AD, PD-AD), further GLMs were constructed containing the data from the disease groups and matched HC group, investigating the same metrics (power, spectral shape, and connectivity). We included a separate regressor for each group and used a two-tailed contrast design matrix (see Supplementary Figure 2). Further included regressors were age, sex, scanner type, and grey matter volume. Grey matter volume was included to account for differences in atrophy between disease groups (Supplementary Material - Grey Matter Volume Extraction).

The positive linear relationship between beta power and age has been well described (Rossiter et al., 2014). To explore this relationship, a further GLM was constructed containing all available participants with group memberships preserved (total n = 278 (PD, n = 25), (ALS, n = 33), (AD, n = 29), (HCs, n = 191)) with age as the regressor to assess for differences in the ageing trajectories of beta power between groups (see Supplementary Figure 3 for design matrix). These differences in trajectories were visualised by taking the mean beta power across parcels and performing a linear regression for each group separately.

For all statistical analyses, we tested the null hypotheses that there were no group differences or no effect of age. Multiple comparisons across regions (N = 52) (and frequencies, N = 6, where relevant) were accounted for using the maximum t-statistic family-wise error rate correction method. T-statistics were calculated on the true dataset and then for 5000 random permutations of the data. By taking the maximum t-statistic across all parcels and frequencies in each permutation, a null distribution was created against which the t-statistic from the true (unshuffled) dataset was compared. The P-values were calculated by computing the proportion of maximum t-statistics from permutations that were smaller than the true t-statistic ($p < 0.05$ significant). All P-values were reported after correction for multiple comparisons. This non-parametric statistical method was chosen as it is robust to differences in group sizes and to non-normal distribution of data (Quinn et al., 2024). When multiple regions showed significant changes for a single metric, a key region was reported to illustrate the effect (full list of significant results in Supplementary Material). Statistical results were reported using the notation (t(degrees of freedom), p) where *degrees of freedom = n_participants – (n_comparisons + n_regressors 1 + (intercept))*. Contrast of Parameter Estimates (copes) were estimated to illustrate the size of effect of each contrast (Quinn et al., 2024).

To concentrate our analysis and interpretation, we chose four previously identified important sub-metrics in neurodegeneration: beta power, 1/f exponent, oscillatory speed (CoE), and gamma connectivity (Boon et al., 2019; Mandal et al., 2018; Rossiter et al., 2014; Trubshaw et al., 2024).

Code for data processing and statistical analyses can be found at: https://github.com/OHBA-analysis/Trubshaw2025_NeurodegenerativeDisorders.

# 3   Results

Participant characteristics are summarised in Table 2. Groups included in the analyses were AD (n = 29), PD (n = 25), ALS (n = 33), age- and sex-matched HCs (n = 114), and a group of younger HCs (n = 77). The significant differences in the ages of group participants were mitigated by including age as a regressor in GLMs. A full table of parcellated Glasser52 brain regions and all significant results can be found in Supplementary Table 2 and Supplementary Material **(full_-results)** respectively.

## 3.1   MEG markers of ageing in the healthy cohort

### 3.1.1   Healthy ageing was associated with increased beta power, reduced 1/f exponent, and increased oscillatory speed:
In the combined HC groups (n = 191), beta power was higher in older age within multiple sensorimotor regions (somatosensory cortex - right: (t(183) = 1.753, p = 0.002), left: (t(183) = 1.680, p = 0.002)) (Fig. 2B). The 1/f exponent was inversely related to age in occipital (right: (t(183) = -0.745, p = 0.004), left: (t(183) = -0.735, p = 0.006)) and temporal regions (right: (t(183) = -0.754, p = 0.003), left: (t (183) = -0.836, p = 0.001)) (Fig. 3A – **top row**). Oscillatory speed increased proportionally to age in superior frontal regions (right: (t (183) = 1.276, p < 0.001), left: (t(183) = 0.985, p = 0.002)) and sensorimotor regions (right: (t(183) = 0.768, p = 0.016), left: (t(183) = 0.950, p = 0.002)) (Fig. 3A – **bottom row**). There were no significant changes in connectivity associated with healthy ageing (Fig. 4B).

Further findings included significantly reduced delta (t(183) = - 1.684, p = 0.002) and theta power (t(183) = -1.978, p = 0.001), reduced alpha power in occipital (t(183) = -2.299, p = 0.001), increased alpha power in temporal regions (t(183) = 1.227, p = 0.032) and increased gamma (t(183) = 1.926, p = 0.001) and high-gamma (t (183) = 1.872, p = 0.001) power in occipital regions with increasing age (Fig. 2B – **rows 2 and 3, right**). See Supplementary Material **(full_results > healthy_ageing_comparison)** for results of all significant statistical tests.

## 3.2   Frequency band spectral power

### 3.2.1   Neurodegenerative diseases were characterised by reduced beta power with increased age, in contrast to the trajectory of the healthy ageing group:
Contrary to the increase in beta power observed in healthy ageing (described above), all three neurodegenerative diseases showed reductions in beta power compared to matched controls (n = 114) in sensorimotor regions (left somatosensory: PD-HC (t(183) = -2.905, p = 0.003), ALS-HC (t(183) = -2.466, p = 0.017), AD-HC (t(183) = - 5.718, p = 0.001)) (Fig. 2A – **row 1, right**). AD showed a larger reduction in beta power than ALS (ALS-AD: left somatosensory: t(183) = 2.780, p = 0.041) and PD (PD-AD: right somatosensory: t (183) = 3.005, p = 0.023). In AD, reductions in beta power were visible across the entire cortex, whereas in PD and ALS these changes were more limited to sensorimotor and inferior frontal regions (Fig. 2A – **row 1, right**). Distinctively, compared to HC, AD showed widely increased delta power (right visual: (t(183) = 7.107, p = 0.001), right somatosensory: (t(183) = 5.170, p = 0.001), left dorsolateral prefrontal: (t(183) = 4.019, p = 0.001)) (Fig. 2A – **row 1, left)** in contrast to few changes observed in PD and

ALS and strong decreases seen in healthy ageing (right somatosensory: (t(183) = -1.574, p = 0.003), left inferior frontal: (t(183) = -1.245, p = 0.025)). It was demonstrated that AD showed significantly greater delta power than both ALS and PD by testing the ALS-AD and PD-AD effects (Fig. 2A **– row 1, left)**.

Contrary to the reduced sensorimotor theta power seen in healthy ageing (right somatosensory: t(183) = -2.414, p = 0.001)), theta power was increased in both PD (PD-HC: right somatosensory: t(183) = 4.323, p = 0.001) and AD (AD-HC: right somatosensory: t(183) = 4.421, p = 0.001) (Fig. 2A **– row 2, left)**.

Compared to controls, PD was uniquely characterised by increased alpha power in parietal regions (right inferior parietal: (t(183) = 2.379, p 0.028), in contrast to marked reduced alpha power seen in AD (right inferior parietal: (t(183) = -2.570, p = 0.018)) and ALS (left temporal: (t(183) = -2.446, p = 0.017)) (Fig. 2A **- row)**. AD showed the largest reductions in alpha power out of the three diseases.

In healthy ageing, only the primary visual cortex showed increased gamma power (t(183) = 1.729, p = 0.002). However, in neurodegen-erative diseases, sensorimotor, rather than occipital, high-gamma power was increased in both AD (left somatosensory: t(183) = 2.547, p = 0.020) and ALS (left somatosensory: t(183) = 2.962, p = 0.001), compared to HC (Fig. 2A **– row 3, right)**.

Supplementary Material **(full_results > group_comparison > power)** shows results from power comparisons from all frequency bands and contrasts in the neurodegenerative diseases.

The positive linear relationship between beta power and age has been previously described (Rempe et al., 2022). The relationship was investigated further by taking the mean beta power from all parcels in each participant and regressing against age separately for each patient group (Supplementary Figure 4A). The younger HC cohort was included in the HC group in this analysis (total HC N = 191). There was an increase in beta power across regions proportional to age in HC (r = 0.463, p < 0.001) (Supplementary Figure 4A). There was no association of age with beta power in AD (r = 0.090, p = 1.000), PD (r = 0.278, p = 0.716), or ALS (r = 0.302, p = 0.349). The regional rather than the whole-brain analysis revealed a reduction in beta power in AD with increasing age in occipital (t(262) = -4.361, p = 0.002) and inferior frontal regions (t(262) = -4.247, p = 0.002) (Supplementary Figure 4Bi). Compared to HC, the effect of age on increasing beta power in the disease groups was reduced in AD (right somatosensory cortex: t (262) = -4.356, p = 0.001), in PD (left premotor: t(262) = -3.462, p = 0.010), and ALS (right somatosensory cortex: t(262) = -4.776, p = 0.001) (Supplementary Figure 4Bii).

For reference, Supplementary Material **(full_results > beta_power_ageing_group_comparison)** contains a full list of significant results for beta power ageing trajectory group comparisons.

### 3.3 Spectral shape

### 3.3.1 Oscillatory slowing was common to all neurodegenerative diseases, contrasting with the oscillatory acceleration in the healthy ageing group. A reduced 1/f exponent was unique to ALS, compared to increases seen in AD and PD

Healthy ageing was associated with oscillatory acceleration (increased CoE) in sensorimotor (right somatosensory: t(183) = 0.768, p = 0.016) and superior frontal regions (left dorsolateral prefrontal cortex: t(183) = 0.985, p = 0.002) (Fig. 3 – **bottom row**). In contrast, compared to controls, AD, PD, and ALS all showed significant oscillatory slowing (reduced CoE) in sensorimotor regions (right somatosensory: AD (t(183) = -1.992, p = 0.010), PD (t(183) = -3.590, p = 0.001), ALS (t(183) = -1.929, p = 0.008)) (Fig. 3 – **bottom row**). Reductions compared to matched controls in oscillatory speed were observed across the entire cortex (rather than just sensorimotor regions) in AD (right visual: (t(183) = -5.721, p = 0.001), left dorsolateral prefrontal (t (183) = -5.397, p = 0.001)) and PD (right visual: (t(183) = -3.191, p = 0.001), left dorsolateral prefrontal (t(183) = -2.836, p = 0.005)) (Fig. 3 – **bottom row**).

There were no significant changes in the alpha or beta peak frequencies in healthy ageing (Supplementary Figure 5A). Therefore, the increase in sensorimotor and frontal CoE observed in healthy ageing was likely explained by a significant increase in beta peak power in sensorimotor (t(183) = 1.906, p = 0.001) and frontal (t(183) = 1.908, p = 0.001) regions (Supplementary Figure 5A). In contrast, all three disease groups showed a shift in alpha peak frequency to the left (slower frequencies) (AD (t(183) = -1.992, p = 0.010), PD (t(183) = -2.684, p = 0.001), ALS (t(183) = -2.730, p = 0.009)) and a drop in beta peak power (AD (t(183) = -5.332, p = 0.001), PD (t(183) = -2.724, p = 0.010), ALS (t(183) = -2.406, p = 0.039)) (Supplementary Figure 5 B). Compared to controls PD and ALS showed increased alpha peak power (inferior parietal cortex: PD (t(183) = 2.721, p = 0.010), ALS (t(183) = -2.542, p = 0.025)) and ALS showed decreased beta peak frequency (posterior cingulate cortex ALS (t(183) -2.764, p 0.011)) whilst AD showed decreased alpha peak power (superior somatosensory: AD (t(183) = 2.857, p = 0.007)) and increased beta peak frequency (inferior frontal: AD (t(183) = 3.681, p = 0.001)) (Supplementary Figure 5 B). AD showed a reduced number of detectable beta peaks compared to controls (t(183) = -9.628, p < 0.001), ALS (t(183) = - 7.682, p < 0.001) and PD (t(183) = -6.477, p < 0.001). There were no other significant differences in the number of detected alpha or beta peaks between groups (p > 0.1).

Healthy ageing was associated with reduced 1/f exponent in occipital (right visual: t(183) = -0.745, p = 0.004) and temporal (right temporal: t(183) = -0.593, p = 0.047) regions (Fig. 3 – **top row**). In contrast, compared to controls, the 1/f exponent increased in both AD (right visual: (t(183) = 3.398, p = 0.001), right temporal: t(183) = 3.417, p = 0.001)) and PD (right visual: t(183) = 3.072, p < 0.001) (Fig. 3 – **top row**). ALS was unique in showing a *reduced* 1/f exponent in occipital (right visual: t(183) = -2.704, p = 0.001), frontal (left dorsolateral prefrontal: t(183) = -2.133, p = 0.001), and temporal (left temporal t(183) = -1.842, p = 0.005) regions relative to HC (Fig. 3 – **top row**).

For reference Supplementary Material **(full_results > group_-comparison > 1 f/coe/peak)** shows spectral shape results from all contrasts.

### 3.4 Connectivity

#### 3.4.1 In contrast to healthy ageing and PD, which showed few connectivity changes, AD and ALS were characterised by severe disruptions in connectivity:
There were no significant ageing effects on global connectivity in HC (p > 0.1) (Fig. 4**B**). AD showed the most connectivity disruptions compared to HC, including increased connectivity across all brain regions in the delta band (right visual: $(t(183) = 3.742, p = 0.002)$, right temporal $(t(183) = 3.274, p = 0.007)$, left dorsolateral prefrontal $(t(183) = 3.332, p = 0.007))$ (Fig. 4**A – row 1, left**), and reduced beta connectivity across all regions (right visual: $(t(183) = -4.164, p = 0.001)$, right temporal $(t(183) = -3.440, p = 0.003)$, left dorsolateral prefrontal $(t(183) -3.814, p\ 0.002))$ (Fig. 4**A – row 1, right**). Alpha connectivity was uniquely reduced in AD across most of the cortex (right temporal: $(t(183) = -4.910, p = 0.001)$, left dorsolateral prefrontal cortex: $(t(183) = -3.319, p = 0.007))$ in contrast to PD and ALS, which showed no significant differences in alpha connectivity (Fig. 4**A – row 3, left**). Both AD and ALS were characterised by increased gamma connectivity compared to HC in temporal (right - AD: $(t(183) = 2.992, p = 0.017)$, ALS: $(t(183) = 2.746, p = 0.026)$, occipital (right visual – AD: $(t(183) = 4.416, p = 0.001)$, ALS: $(t(183) = 2.961, p = 0.015)$, and frontal (left dorsolateral prefrontal – AD: $(t(183) = 2.738, p = 0.028)$; ALS: $(t(183) = 2.669, p = 0.031)$ regions (Fig. 4**A – row 2, right**).

For reference, Supplementary Material **(full_results > group_-comparison > aec)** shows connectivity results from all frequency bands and contrasts.

## 4 Discussion

### 4.1 Divergent spectral profiles – ageing and disease

All three neurodegenerative disease groups (AD, PD, and ALS) were associated with a common reduction in beta power and oscillatory slowing (CoE), contrasting with the increases in these metrics seen with older age in the healthy cohort. The most severe and widespread disruptions to cortical neurophysiology were observed in people with AD. The power spectrum slope was uniquely reduced in ALS. Spectral power changes relative to healthy controls in the neurodegenerative disorder groups were opposite to those observed in healthy ageing in the delta, theta, alpha and beta frequency bands. The 1/f exponent (which may at least partly reflect excitatory:inhibitory balance (Donoghue et al., 2020; Gao et al., 2017; Muthukumaraswamy and Liley, 2018)) showed opposite directionality to healthy ageing (Gerster et al., 2022). It is theorised that many of the neurophysiological changes seen in healthy ageing represent compensatory or adaptive responses (López et al., 2014). While such changes might equally represent the start of a decompensation phase, in either case, the cortical neurophysiology associated with neurodegeneration does not appear to be an exaggerated form of 'natural ageing'.

### 4.2 Concordant spectral profiles – ageing and disease

Only in high-gamma power did both healthy ageing and neurode-generative diseases (AD and ALS, but not PD) show the same directionality (increased power) (Trubshaw et al., 2025; van Deursen et al., 2008). Increases in high-gamma may relate to disruption of GABAergic interneuron circuits, known to become dysfunctional in healthy ageing and neurodegeneration (Ambrad Giovannetti and Fuhrmann, 2019; Jafari et al., 2020; Kujala et al., 2015; Lozovaya et al., 2018; Rozycka and Liguz-Lecznar, 2017; Turner and Kiernan, 2012). PD did not, however, show increased high-gamma power compared to HC, potentially due to its relative sparing of cortical interneurons compared to AD and ALS (Cherian et al., 2024).

### 4.3 Findings in healthy ageing

Corroborating our work, healthy ageing has been associated with reductions in low frequency (delta-alpha) power (Beese et al., 2017; Cummins and Finnigan, 2007; Rempe et al., 2023; Vlahou et al., 2014) and increases in beta power (Dustman et al., 1999; Gómez et al., 2013; Heinrichs-Graham et al., 2018; Heinrichs-Graham and Wilson, 2016; Hübner et al., 2018; Koyama et al., 1997, p. 199; Stier et al., 2025; Veldhuizen et al., 1993). Slowing of the alpha peak frequency with age (Cesnaite et al., 2023; Dustman et al., 1993; Gohil et al., 2024; Sahoo et al., 2020; Scally et al., 2018) has been negatively correlated to cognitive performance (Cesnaite et al., 2023; Gohil et al., 2024), although this finding was not observed in the current dataset. Increased connectivity in the alpha and beta bands correlated with better performance in healthy ageing, suggesting compensatory responses (Geerligs et al., 2014). Our study identified a marked reduction in beta connectivity in AD (Cuesta et al., 2015; Gómez et al., 2018; López-Sanz et al., 2016). The reductions seen in the PD and ALS groups were not significant, though previous studies have reported changes in this direction (Boon et al., 2020; Dukic et al., 2019; Kohl et al., 2024). If increasing beta connectivity is a marker of compensation, reduced beta connectivity might therefore represent a failure of compensatory processes.

### 4.4 Unique profiles of disease

The heterogeneous but overlapping symptomatology and pathophysiology profiles of AD, PD, and ALS have led to debate over whether neurodegenerative diseases represent discrete disorders or a continuum (Armstrong, 2012). Certainly, vast neurophysiological differences between disease groups exist.

This study found that PD showed increased parietal alpha power, in contrast with reduced alpha power seen in ALS and AD (Dukic et al., 2019; Lejko et al., 2020; Zhu et al., 2019). Alpha power results in PD have previously been heterogeneous, which may reflect a shift from increased to decreased power across the disease course (Boon et al., 2020). The increased alpha connectivity we observed in PD has also been previously described (Boon et al., 2019). Reduced alpha power has been well described in AD (Huang et al., 2000; Penttiläet al., 1985; Schreiter-Gasser et al., 1993; Stam et al., 2005; Wang et al., 2015, p. 20) and may relate to damage to the cholinergic ascending system from basal forebrain to the cortex (Babiloni et al., 2021; Meghdadi et al., 2021). Of the disease groups, AD uniquely showed increased delta power (de Haan et al., 2008; Fernández et al., 2013, 2006, 2003,

2002; Huang et al., 2000; Penttiläet al., 1985; Schreiter-Gasser et al., 1993; Wang et al., 2015) and reduced alpha connectivity, which might support the concept of a 'disconnection syndrome' with loss of functional 'hub regions' (Elgandelwar and Bairagi, 2021; Koelewijn et al., 2017; Schoonhoven et al., 2022).

ALS was uniquely characterised by increased signal complexity (reduced 1/f exponent), in contrast with reduced signal complexity (increased 1/f exponent) in PD and AD (Helson et al., 2023; Sun et al., 2020; Trubshaw et al., 2024). This increased signal complexity in ALS, previously reported (Trubshaw et al., 2024), may reflect cortical hyperexcitability, a consistent pathophysiological feature of the symptomatic (Menon et al., 2015) and peri-symptomatic (Vucic et al., 2008) disease phases. Consistent with our findings in symptomatic AD, prior work found a shift from reduced 1/f in prodromal (hyperexcitable) to increased 1/f hypoexcitability in later stages (Martínez-Cañada et al., 2023). PD has consistently shown increased 1/f exponent, perhaps related to increased GABAergic, NMDA, or decreased glutamatergic activity (Helson et al., 2023; Wiesman et al., 2024, 2022).

### 4.5 Similarities between diseases

Reductions in beta power and oscillatory slowing stand out as potential common biomarkers of neurodegeneration and confirm that these diseases show strong neurophysiological similarities. Oscillatory slowing has been well documented in AD (de Haan et al., 2008; Fernández et al., 2013, 2006, 2003, 2002; Huang et al., 2000; Penttilä et al., 1985; Schreiter-Gasser et al., 1993; Wang et al., 2015) and PD (Boon et al., 2023; Bosboom et al., 2006; Krösche et al., 2023a; Neufeld et al., 1988; Olde Dubbelink et al., 2013; Soikkeli et al., 1991; Stanzione et al., 1996; Stoffers et al., 2007; Wiesman et al., 2023), posited to reflect neuronal dysfunction and protein aggregate deposition (Bruña et al., 2023; Coomans et al., 2021; Jafari et al., 2020; Krösche et al., 2023a; Stoffers et al., 2007; Wang et al., 2015) but not previously reported in ALS. The peak analysis in this work showed that oscillatory slowing in neurodegeneration is driven by a shift of the alpha peak to the left (Babiloni et al., 2017; Benwell et al., 2020; Boon et al., 2023; Meghdadi et al., 2021; Montez et al., 2009; Olde Dubbelink et al., 2013; Poza et al., 2007; Puttaert et al., 2021; Soikkeli et al., 1991), and a reduction in beta peak power. In the AD group, the reduction in beta peak power was so pronounced that significantly fewer beta peaks were detected compared to all other groups. The increase in oscillatory speed in healthy ageing was likely caused by increased beta peak power (Azami et al., 2023; Krösche et al., 2023b). Beta power is governed by a complex interplay of thalamo-cortical signalling, GABAergic inter-neuronal activity, and other local neuronal circuitry (Cheyne, 2013; Reis et al., 2019; Sherman et al., 2016; Shin et al., 2017). Beta power, with its integral role in movement, might be expected to be disrupted in PD and ALS, both of which have strong movement components (Cheyne, 2013; Yoganathan et al., 2025). In AD, the cause for reduced beta is less clear, although it has previously been correlated with burden of CSF tau, suggesting reduced beta may well represent cortical neuronal loss (Smailovic et al., 2018). Beta likely plays a broader role in integrating sensory inputs with cognitive and motor planning circuits (disrupted in AD), rather than simply linearly 'causing' movement (Barone and Rossiter, 2021; Engel and Fries, 2010; Hallett et al., 2021). Conversely, other MEG metrics (like alpha power (Babiloni

et al., 2021)) may be more sensitive to the underlying neurobiology governing cognitive impairment, with AD showing the most disrupted neurophysiology, compared to PD and ALS, in which cognitive changes are usually milder or limited to late-stage disease. Work is required to investigate this hypothesis further.

### 4.6 Limitations

The statistical significance testing employed in this work used strict multiple comparisons correction and accounted for confounding factors (see Supplementary Figures 1 and 2). This minimised the risk of Type I errors. It should be noted that challenges exist in applying the FOOOF algorithm to certain PSDs, in particular if they contain spectral plateaus and overlapping oscillatory components (Gerster et al., 2022). However, the PSD of MEG data is particularly suited to the application of FOOOF (Gerster et al., 2022) and we optimised parameters (see 'Metric calculation') to ensure robust estimates. Further limitations of the current study include the lack of longitudinal data, which might help solidify metrics such as beta power and oscillatory slowing as biomarker targets. Future work might seek to include longitudinal measurements, together with risk groups, such as people with rapid-eye-movement sleep behaviour disorder (high risk for PD) or carriers of high-risk genotypes in AD (*APOE*) and ALS (*C9orf72*) (Foraker et al., 2015; Talbot et al., 2018; Trubshaw et al., 2025a). Future work might focus on implementing dynamic brain network analyses that have more interpretability and sensitivity to subtle neurophysiological changes (Gohil et al., 2022; Metzger et al., 2024).

## 5 Conclusions

Our findings show that MEG can identify distinct patterns of cortical neurophysiology associated with common neurodegenerative disease states and how they diverge from healthy ageing. Oscillatory slowing and beta power are leading potential biomarkers for pre-symptomatic neurodegenerative disorders, with the reduced 1/f exponent worthy of more focused study as potentially specific to ALS. This is a small step towards achieving the long-term goal of preventative medicine for these disorders.

## Supplementary Material

Refer to Web version on PubMed Central for supplementary material.

## Acknowledgments

The authors would like to thank Anna Camera and Sven Braeutigam for their help with acquiring the data and all participants for their time and enthusiasm. This research was funded by the Marie Skłodowska-Curie Innovative Training Network "European School of Network Neuroscience (euSNN)" (860563), a Wellcome Trust Senior Investigator Award to A.C.N. (104571/Z/14/Z), and a James S. McDonnell Foundation Understanding Human Cognition Collaborative Award (220020448). The Wellcome Centre for Integrative Neuroimaging is supported by core funding from the Wellcome Trust (203139/Z/16/Z and 203139/A/16/Z). This research was supported by the NIHR Oxford Health Biomedical Research Centre (NIHR203316). The views expressed are those of the author(s) and not necessarily those of the NIHR or the Department of Health and Social Care. For the purpose of open access, the author has applied a CC BY public copyright licence to any Author Accepted Manuscript version arising from this submission. The OPDC Discovery cohort is funded by Parkinson's UK and the Oxford NIHR Biomedical Research Centre. MW's research is additionally supported by the Wellcome Trust (106183/Z/14/Z, 215573/Z/19/Z), the New Therapeutics in Alzheimer's Diseases (NTAD) study supported by UK MRC, the Dementia Platform UK (RG94383/RG89702). This work has been funded by the Medical Research Council (MC_UU_00030/14;

MR/T033371/1; the Dementia Platform UK), the Wellcome Trust (220258); and the NIHR Cambridge Biomedical Research Centre (NIHR203312). The views expressed are those of the authors and not necessarily those of the NIHR or the Department of Health and Social Care. For the purpose of open access, the authors have applied a CC BY public copyright licence to any Author Accepted Manuscript version arising from this submission.

## Data availability

Data will be made available on request.

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

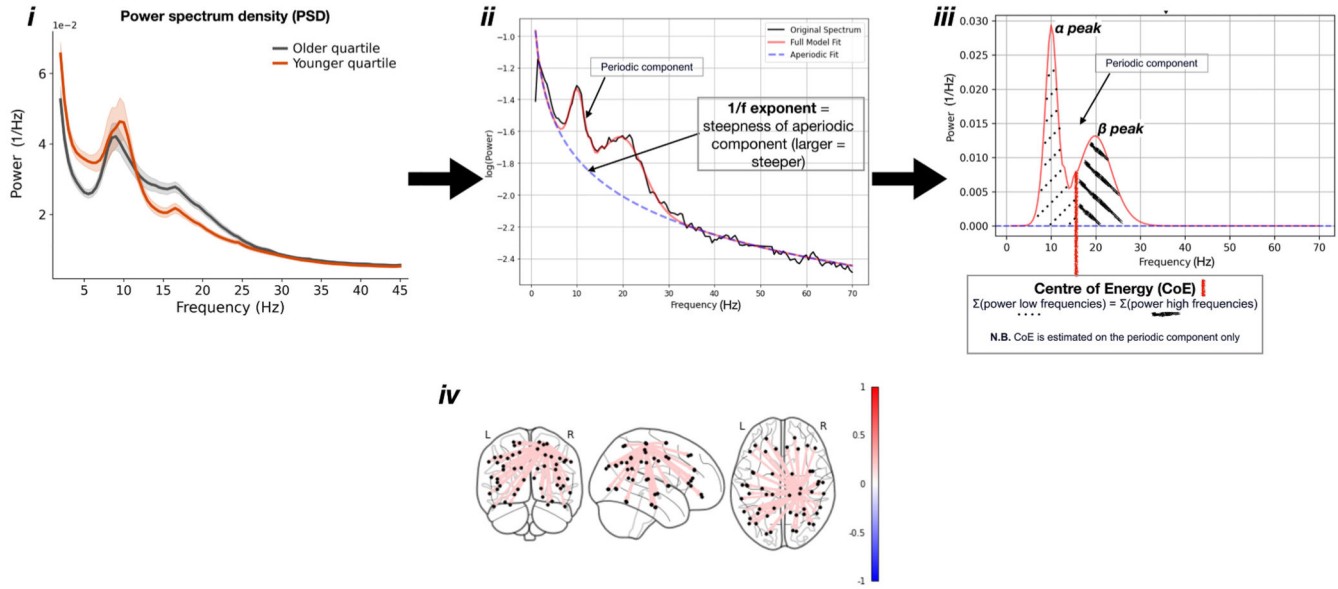

**Fig. 1. Spectral measures estimation.**

*i* - shows an example power spectrum density (PSD) of the youngest quartile (mean age 29.2 years, red), versus the oldest quartile (mean age 74.8 years, grey) of the extended healthy control group (n = 191). This illustrates that with older age, there is a general flattening of the PSD, beta power increases and theta power decreases. *ii* – illustrates the estimation of the 1/f exponent using the FOOOF algorithm. The 1/f exponent represents the steepness of the slope of the aperiodic fit (dashed blue line). A larger 1/f exponent reflects a steeper slope (reduced signal complexity). *iii* – illustrates the estimation of oscillatory slowing (Centre of Energy, CoE). The aperiodic component is extracted from the full PSD (de-FOOOFed spectrum). The CoE is the frequency at which the summated power of lower frequencies is equal to the summated power of higher frequencies. *iv* – illustrates the estimation of connectivity. The amplitude envelope correlation (connectivity) of each timecourse is calculated pairwise. For each region, the mean value across one axis of the connectivity matrix is calculated and represents an estimate of each region's global connectivity.

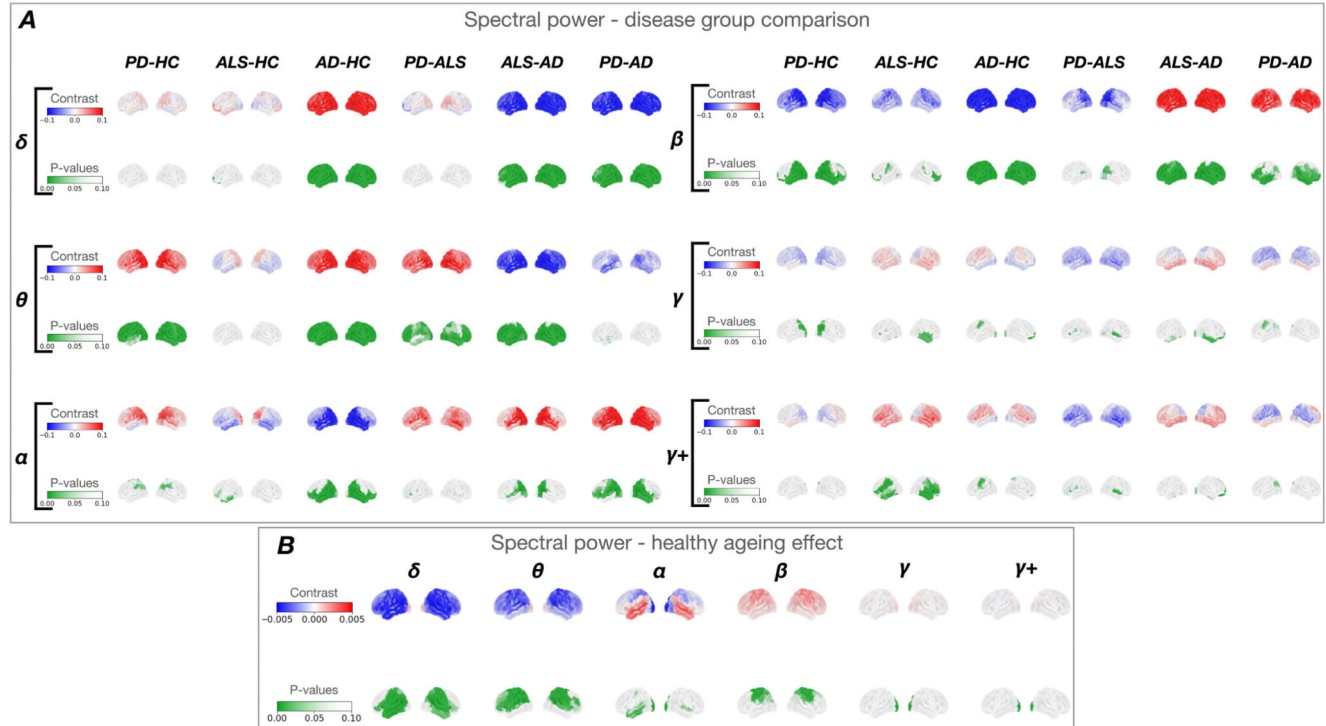

**Fig. 2. Spectral power results.**
*A* – power in disease groups. Power differences in each canonical frequency band between each group of participants (Alzheimer's Disease (AD), n = 29; Parkinson's Disease (PD), n = 25, amyotrophic lateral sclerosis (ALS), n = 33 and age-matched HC, n = 114). *B* – power in healthy ageing. Power changes associated with healthy ageing in the healthy control group (n = 191) with the effect of age as a continuous regressor. The top row in each frequency band represents the effect of age, or the difference between groups. Effects (contrasts) are shown on a scale of blue to red, with red representing a positive and blue a negative effect. The bottom row represents the P-value map with p < 0.1 shown in green. Notably, contrary to healthy ageing, beta power was reduced across all neurodegenerative disorders with the largest and most widespread reduction seen in AD and the smallest and most restricted topographically in ALS. Both PD and AD showed increases in theta power, contrary to the decrease seen with healthy ageing. ALS and AD showed decreases in alpha power but increases in high-gamma power in fronto-temporal (ALS only) and motor regions. Gamma power increased in healthy ageing in occipital regions which was not seen in any of the neuro-degenerative diseases.

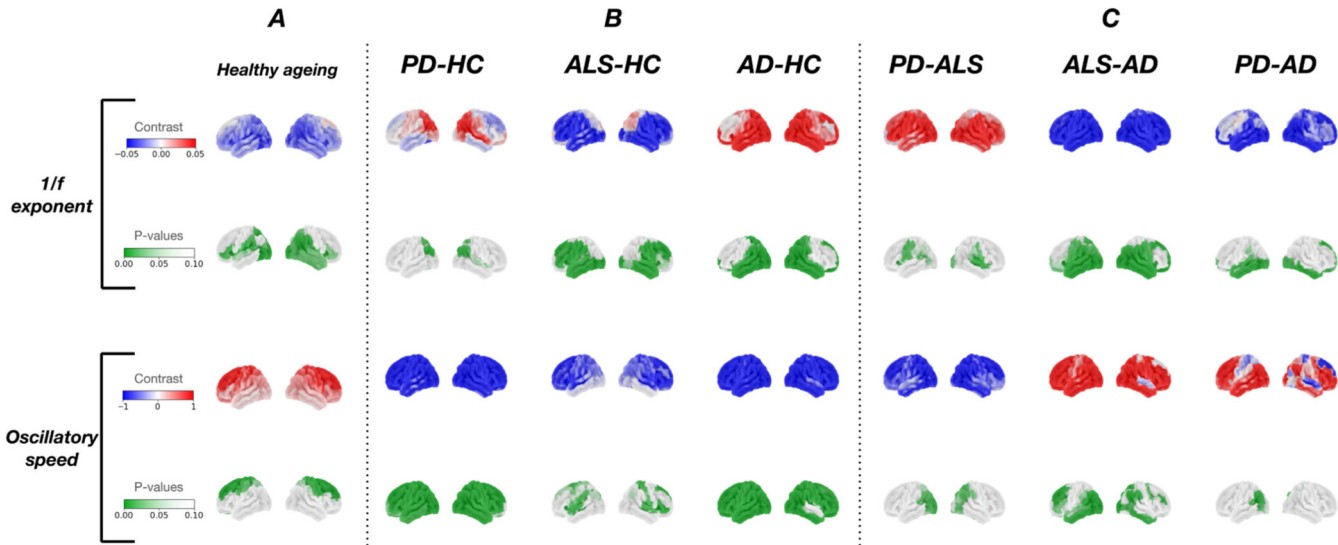

**Fig. 3. Spectral shape results.**

Spectral shape changes associated with *A* - healthy ageing in a large group of healthy controls (HC, n = 191). *B* - disease group comparisons with age-matched healthy controls (Alzheimer's Disease (AD), n = 29; Parkinson's Disease (PD), n = 25, amyotrophic lateral sclerosis (ALS), n = 33 and age-matched HC, n = 114). **C** - comparisons between neurodegenerative disease groups. The top row in each metric represents the effect of age, or the difference between groups. Effects (contrasts) are shown on a scale of blue to red, with red representing a positive and blue a negative effect. The bottom row represents the P-value map with p < 0.1 shown in green. The 1/f exponent was reduced in both ALS and in healthy ageing, representing increased signal complexity, contrary to both PD and AD. Contrary to the increase in oscillatory speed associated with healthy ageing in superior frontal and motor regions, all three neurodegenerative diseases showed a reduction in oscillatory speed when compared to HC. AD showed the slowest oscillatory speed of the three diseases.

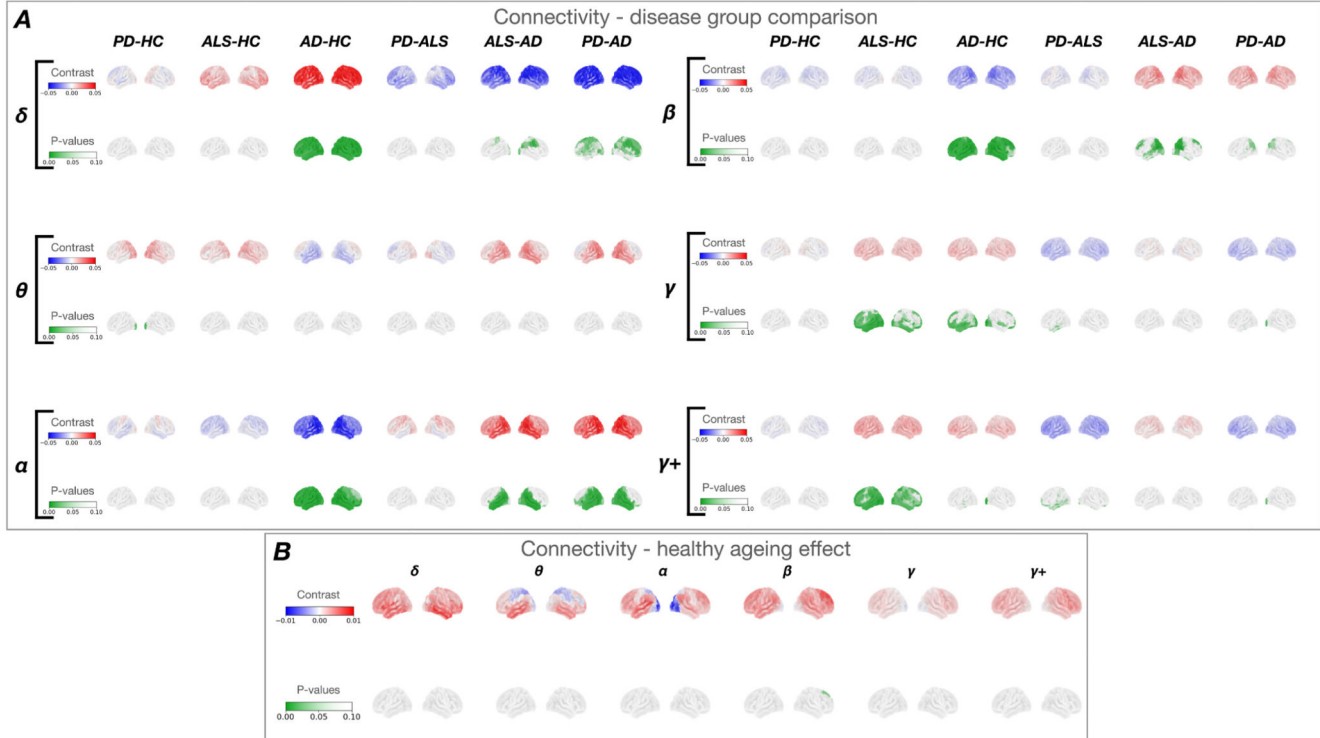

**Fig. 4. Connectivity results.**

*A* – connectivity in disease groups. Connectivity differences in each canonical frequency band between each group of participants (Alzheimer's Disease (AD), n = 29; Parkinson's Disease (PD), n = 25, amyotrophic lateral sclerosis (ALS), n = 33 and age-matched HC, n = 114). *B* – connectivity in healthy ageing. Connectivity changes associated with healthy ageing in the healthy control group (n = 191) with the effect of age as a continuous regressor. The top row in each frequency band represents the effect of age, or the difference between groups. Effects (contrasts) are shown on a scale of blue to red, with red representing a positive and blue a negative effect. The bottom row represents the P-value map with p < 0.1 shown in green. Notably, no significant changes in connectivity were seen in healthy ageing. In contrast, ALS and AD showed an increase in gamma connectivity compared to controls. AD showed a reduction in alpha and beta but increase in delta connectivity.

**Table 1 Summary table of core features of three neurodegenerative diseases.**

|  | AD | PD | ALS |
|---|---|---|---|
| **Motor symptoms** | Minimal | Bradykinesia, tremor, rigidity, postural instability | Progressive neuromuscular weakness |
| **Cognitive symptoms** | Memory, executive function | Memory, executive function | Executive function |
| **Psychological symptoms** | Apathy, depression | Apathy, depression | Apathy, disinhibition, loss of social cognition |
| **Median survival from symptom onset (standardised 60 yrs)** | 8.8 years | 14.6 years | 30 months |
| **Core proteinopathy** | amyloid-β, tau | α-synuclein | TDP-43 |

**Table 2 Summary table of participant demographics.**

| | AD | PD | ALS | Age- matched HC (HCo) | Young HC (HCy) |
|---|---|---|---|---|---|
| **Number** | 29 | 25 | 33 | 114 | 77 |
| **Mean age* (SD)** | 72.1 (8.0) | 68.0 (6.0) | 62.8 (10) | 63.8 (10.4) | 46.9 (22.7) |
| **Age distribution (quartile 1, quartile 3 (interquartile range)** | 68.3, 78.5 (10.2) | 65.0, 72.0 (7.0) | 53.0, 70.0 (17.0) | 57.5, 70.6 (13.3) | 24.0, 69.0 (45.0) |
| **% male^** | 48.3 | 60.0 | 57.6 | 57.9 | 49.3 |
| **Disease score** | Addenbrooke's Cognitive Assessment (ACE) (0–100 with lower scores reflecting more cognitive impairment) | Unified Parkinson's Disease Rating Scale – Motor score (UPDRS-Motor) (0, 108 with higher scores reflecting more severe motor impairment) | ALS Functional Rating Score – Revised (ALSFRS-R) (0–48 with lower scores reflecting more disability) | - | - |
| **Mean disease score (SD)** | 67.7 (12.4) | 29.8 (10.4) | 37.8 (6.3) | - | - |
| **Mean years from symptom onset (SD)** | 4.1 (2.3) | 2.8 (1.5) | 2.0 (1.3) | - | - |

*GLM results comparing age distributions between groups:

| | t-statistics | P-value (* significant) |
|---|---|---|
| HCo-HCy | 7.82 | *<0.01 |
| HCo-PD | -1.30 | *0.05 |
| HCo-ALS | 0.37 | 0.55 |
| HCo-AD | -2.73 | *<0.01 |
| HCy-PD | -6.25 | *<0.01 |
| HCy-ALS | -5.19 | *<0.01 |
| HCy-AD | -7.90 | *<0.01 |
| PD-ALS | 1.36 | 0.12 |
| PD-AD | -1.03 | 0.25 |
| ALS-AD | -2.52 | *<0.01 |

^Chi-squared test found no statistically significant difference in the proportion of males in each group: $\chi^2$ = 2.23, adjusted P-value = 1.00, degrees of freedom = 4,

