## [Peer Review File · Progress in neurobiology]

Peer Review Overview

Manuscript Title: "Divergence of cortical neurophysiology across different neurodegenerative disorders compared to healthy ageing"

Received	27-Mar-2025
1 st Decision	09-Jun-2025
Revision Submitted	18-Sep-2025
Accepted	02-Dec-2025

Decision Letter

Dear Dr. Trubshaw,

Thank you for submitting your manuscript to Progress in Neurobiology.

We have completed our evaluation of your manuscript. The reviewers recommend reconsideration of your manuscript following major revision. We invite you to resubmit your manuscript after addressing the comments below. Please resubmit your revised manuscript by **Sep 09, 2025**.

When revising your manuscript, please consider all issues mentioned in the reviewers' comments carefully: please outline every change made in response to their comments and provide suitable rebuttals for any comments not addressed. Please note that your revised submission may need to be re-reviewed.

To submit your revised manuscript, please log in as an author at <https://www.editorialmanager.com/proneu/>, and navigate to the "Submissions Needing Revision" folder.

Research Elements (optional)

This journal encourages you to share research objects - including your raw data, methods, protocols, software, hardware and more – which support your original research article in a Research Elements journal. Research Elements are open access, multidisciplinary, peer-reviewed journals which make the objects associated with your research more discoverable, trustworthy and promote replicability and reproducibility. As open access journals, there may be an Article Publishing Charge if your paper is accepted for publication. Find out more about the Research Elements journals at https://www.elsevier.com/authors/tools-and-resources/research-elements-journals?dgcid=ec_em_research_elements_email.

Progress in Neurobiology values your contribution and we look forward to receiving your revised manuscript.

Kind regards,

Brett Foster, PhD
Associate Editor

Jeannie Chin, PhD
Co-Editor-in-Chief
Progress in Neurobiology

Editor and Reviewer comments:

Reviewer #1: This is an interesting paper making use of a large amount of MEG data to look at signatures that characterize neurodegenerative disorders as compared to healthy aging. The paper is important as it follows other non-invasive investigations which strive to identify biomarkers in key populations.

My expertise is methodological. The paper just needs to tidy up on some methodological detail, and possibly add a little information that will generalize across methods.

1. Could the authors please specify the units for power throughout (especially in the figures, i.e. avoid 'arbitrary units'). This would also involve clarifying how the initial beamformer computation is made. (is it normalized in any way. (eg pseudo Z etc)). In other words, what do the spectra in the parcels look like before foof- do they get larger with depth (i.e. not normalized) or are they somehow normalized (and if so by what). This is important as spectral power is the main thing that is quantified (currently). If there is no normalization (eg. <https://www.sciencedirect.com/science/article/pii/S1053811913012305>) then adding the units should be straightforward.

2. Many of the findings hinge on the FOOF algorithm. This is fine, and indeed it is used in many studies, but the authors need to draw attention as to how it may have affected the results (<https://link.springer.com/article/10.1007/s12021-022-09581-8#Sec11> for example).

3. In the metric calculation- it would be useful to include measures that will not be affected by FOOF. For example, the frequency of the alpha and beta peaks will presumably be unaffected (and could be added) but also it would be useful to know the oscillatory speed before foof was applied. This would help future proof this work and make it all the more accessible. This (frequency information) would be especially useful as it would allow the authors to place their findings within a greater body of literature.

4. Is the lack of connectivity (for example, beta connectivity) simply due to lack of power. i.e. if power in a region decreases, won't its connectivity ? (or if some analysis has been done to test for this please expand).

5. In the limitations would certainly include foof. Also any other factors- such as posture or relative head movements of different participant cohorts- that the authors can think of.

Minor.

In the data processing section the authors filter to 125Hz and then resample to 250Hz. The statement 'allowed for studying oscillatory activity up to 125Hz' could be misleading as the filter fall-off is finite and a lot of noise will be aliased into the higher frequency range. Would just remove this statement, as the authors only make use of data up to 80Hz.

In metric calculation section. Maybe make it clearer that the foof etc was applied to each parcel of data. Also include whether this estimate of current flow was normalized (see above).

What are the units of beta power in the regression figure (effect of age) ? Please put real units on both axes throughout and ideally use lines that can be distinguished by colour blind or those of us with b/w printers.

Reviewer #2:

Editorial note: Despite our open peer review, this reviewer explicitly wished for their review to not be disclosed.

Author Response Letter

We thank the reviewers for their time in providing very constructive comments. We respond to each point below.

Reviewer #1

1. Could the authors please specify the units for power throughout (especially in the figures, i.e. avoid 'arbitrary units'). This would also involve clarifying how the initial beamformer computation is made. (is it normalized in any way. (eg pseudo Z etc)). In other words, what do the spectra in the parcels look like before fofof- do they get larger with depth (i.e. not normalized) or are they somehow normalized (and if so by what). This is important as spectral power is the main thing that is quantified (currently). If there is no normalization (eg. <https://www.sciencedirect.com/science/article/pii/S1053811913012305>) then adding the units should be straightforward.

Response: We used a 'unit noise gain' beamformer which normalises the beamformer weights to ensure the source data has unit variance. We used this type of beamformer to control for the depth bias, which means deeper parcels do not have larger PSDs simply due to being deeper. The normalisation in the beamformer results in the source data having no standard unit. Furthermore, after parcellation we standardised (z-transformed) the data, which is another operation that results in unitless data. The PSD of the standardised parcel data has units 1/Hz. We have corrected this in all of the plots.

2. Many of the findings hinge on the FOOF algorithm. This is fine, and indeed it is used in many studies, but the authors need to draw attention as to how it may have affected the results (<https://link.springer.com/article/10.1007/s12021-022-09581-8#Sec11> for example).

Response: In this work, the main use of FOOF was to extract the 1/f exponent for each parcel. Therefore, the main way in which the FOOF algorithm could have affected the results is via a corrupted estimate of the 1/f exponent. The FOOF algorithm is sensitive to certain features in the PSD (see the limitations of FOOF in point 5 below). The PSD of MEG data is particularly suited to FOOF analysis. While the presence of a spectral plateau might potentially influence 1/f exponent estimation, we chose the frequency range for

FOOOF to minimise any estimation bias arising. This is stated in the ‘Metric calculation’ section of the manuscript. We also compared the $1/f$ exponent across groups (taking the difference) rather than studying the value of the $1/f$ exponent directly. This mitigated any systematic biases in the estimation (i.e., errors would ‘cancel out’). Therefore, we consider the results based on the $1/f$ exponent differences to be robust and have explained this in the Discussion section.

3. In the metric calculation- it would be useful to include measures that will not be affected by FOOOF. For example, the frequency of the alpha and beta peaks will presumably be unaffected (and could be added) but also it would be useful to know the oscillatory speed before foof was applied. This would help future proof this work and make it all the more accessible. This (frequency information) would be especially useful as it would allow the authors to place their findings within a greater body of literature.

Response: Including the peak frequencies is an excellent suggestion, and we have now included this as a supplementary analysis. Results can be found in the main text and Supplementary Figure 6.

The most common approach for estimating peak frequencies (centre frequencies) in PSDs is to fit a parametric function to the PSD. This is what FOOOF does (with a particular algorithm designed for robustness). The parametric approach overcomes limitations with the FOOOF-free approach of simply taking the max value in a particular frequency band, which would be sensitive to the frequency resolution of the PSD. In the FOOOF analysis, we did estimate the peak frequencies but did not report them. The method is described in the ‘Metric calculation’ section.

4. Is the lack of connectivity (for example, beta connectivity) simply due to lack of power. i.e. if power in a region decreases, won't its connectivity ? (or if some analysis has been done to test for this please expand).

Response: Thank you. This is an important point to consider. We used the amplitude envelope correlation (AEC) for our measure of connectivity. This is the Pearson correlation between amplitude time courses. The Pearson correlation is an amplitude-normalised measure, which

means it is not sensitive to the amplitude itself, i.e., regions with higher amplitudes will not necessarily have higher connectivity. We have clarified this in the ‘Metric calculation’ section.

5. In the limitations would certainly include foof. Also any other factors- such as posture or relative head movements of different participant cohorts- that the authors can think of.

Response: Agreed. This is now included in the Discussion section.

6. In the data processing section the authors filter to 125Hz and then resample to 250Hz. The statement 'allowed for studying oscillatory activity up tp 125Hz' could be misleading as the filter fall-off is finite and a lot of noise will be aliased into the higher frequency range. Would just remove this statement, as the authors only make use of data up to 80Hz.

Response: This has been removed.

7. In metric calculation section. Maybe make it clearer that the foof etc was applied to each parcel of data. Also include whether this estimate of current flow was normalized (see above).

Response: We have now clarified the measures that are calculated for each parcel individually in ‘Metric calculation’ and clarified the normalisation in ‘Data processing’.

8. What are the units of beta power in the regression figure (effect of age) ? Please put real units on both axes throughout and ideally use lines that can be distinguished by colour blind or those of us with b/w printers.

Response: Due to the normalisation (in both the beamformer and z-transformation of the parcel time courses), the power in the beta band is unitless. We have corrected the units in the plots and have also updated the plots to be colourblind and monochromatic printer friendly.

Reviewer #2

Editorial note: Since this review cannot be published, we also omit the authors' response here.